# qSOFA is a Poor Predictor of Short-Term Mortality in All Patients: A Systematic Review of 410,000 Patients

**DOI:** 10.3390/jcm8010061

**Published:** 2019-01-08

**Authors:** Ronson S. L. Lo, Ling Yan Leung, Mikkel Brabrand, Chun Yu Yeung, Suet Yi Chan, Cherry C. Y. Lam, Kevin K. C. Hung, Colin A. Graham

**Affiliations:** 1Accident and Emergency Medicine Academic Unit, Chinese University of Hong Kong, Hong Kong, China; ronsonsllo@cuhk.edu.hk (R.S.L.L.); lingleung@cuhk.edu.hk (L.Y.L.); Mikkel.Brabrand@rsyd.dk (M.B.); gregory_ycyg@yahoo.com.hk (C.Y.Y.); chan_syi@hotmail.com (S.Y.C.); 1155108119@link.cuhk.edu.hk (C.C.Y.L.); kevin.hung@cuhk.edu.hk (K.K.C.H.); 2Department of Emergency Medicine, Hospital of South West Denmark, Finsensgade 35, DK-6700 Esbjerg, Denmark

**Keywords:** sepsis, qSOFA, prognosis

## Abstract

Background: To determine the validity of the Quick Sepsis-Related Organ Failure Assessment (qSOFA) in the prediction of outcome (in-hospital and 1-month mortality, intensive care unit (ICU) admission, and hospital and ICU length of stay) in adult patients with or without suspected infections where qSOFA was calculated and reported; Methods: Cochrane Central of Controlled trials, EMBASE, BIOSIS, OVID MEDLINE, OVID Nursing Database, and the Joanna Briggs Institute EBP Database were the main databases searched. All studies published until 12 April 2018 were considered. All studies except case series, case reports, and conference abstracts were considered. Studies that included patients with neutropenic fever exclusively were excluded. Results: The median AUROC for in-hospital mortality (27 studies with 380,920 patients) was 0.68 (a range of 0.55 to 0.82). A meta-analysis of 377,623 subjects showed a polled AUROC of 0.68 (0.65 to 0.71); however, it also confirmed high heterogeneity among studies (I^2^ = 98.8%, 95%CI 98.6 to 99.0). The median sensitivity and specificity for in-hospital mortality (24 studies with 118,051 patients) was 0.52 (range 0.16 to 0.98) and 0.81 (0.19 to 0.97), respectively. Median positive and negative predictive values were 0.2 (range 0.07 to 0.38) and 0.94 (0.85 to 0.99), respectively.

## 1. Introduction

Sepsis has been the focus of intensive research efforts over many years, with good reason [1]. Mortality is high (as high as 28.6% [2]) and treatment is expensive ($18,600 USD per hospital stay in the US [3]).

The first international consensus definition of sepsis dates from 1992 [4,5]. It was not substantially updated until 2016 [6] when the task group for the third international consensus definition for sepsis and septic shock redefined sepsis as a “life-threatening organ dysfunction caused by a dysregulated host response to infection” [6] Alongside with this updated definition, the task group also proposed a novel score to identify patients at risk for sepsis: the Quick Sepsis-Related Organ Failure Assessment (qSOFA). However, like many changes, qSOFA has been controversial [7,8,9].

qSOFA was based on the Sepsis-related Organ Failure Assessment (SOFA) score. The SOFA score was originally developed as a predictor for intensive care unit (ICU) mortality [10], and it consists of both vital signs (respiratory rate and blood pressure) and laboratory assessments (liver function tests, urea and creatinine) [6]. qSOFA was intended for use in patients with suspected infection outside of the ICU setting, and included altered mentation, tachypnea, and hypotension [6].

Prior systematic reviews on the topic tend to focus on patients that have already been identified as having suspected infections, which is how the test was originally designed. However, in an Emergency department (ED), the cause for attendance is not always clear, and a diagnosis of infection is often made much later. We there believe that qSOFA should be applied earlier in the treatment process, before a specific condition is considered. This systematic review aims to determine the validity of qSOFA in the prediction of mortality in all patients, with or without a suspected infection.

Objectives: This systematic review examines the validity of qSOFA in predicting in-hospital mortality and 28/30-days mortality, and determines if qSOFA is able to predict ICU admission, length of ICU stay, length of hospital stay, and diagnosis of sepsis, in patients not already identified with a specific condition.

## 2. Methodology

We designed our systematic review using the framework set out in the Preferred Reporting Items for Systematic Reviews and Meta-Analysis Protocols (PRISMA-P) 2015 statement developed with elements adapted from the Cochrane Handbook for Systematic Reviews of Interventions [11,12]. The review was registered with PROSPERO (ID CRD42017063976).

### 2.1. Eligibility Criteria

Types of studies: We considered studies of all designs, except for case series and case reports, i.e., all retrospective and prospective, and all observational and interventional studies. Studies only reported as abstracts were excluded.

Types of participants: All studies with adult patients with or without suspected or confirmed infection, sepsis, severe sepsis, and septic shock were considered. Studies that only included patients with neutropenic fever were excluded from this systematic review, due to the specific nature of this patient group.

Interventions: We considered all studies that reported qSOFA.

Setting: We found studies including patients presenting acutely to Emergency departments and pre-hospital emergency care providers, critical care units (intensive care units and high dependency units), and general wards.

Types of outcome: In-hospital mortality, 1-month mortality, ICU admission, diagnosis of sepsis, length of ICU stay, and length of hospital stay.

Timing: Both retrospective and prospective studies were considered.

Period of review: All studies published until 12 April 2018 were included.

Language: We included articles in languages that the author group could understand (English, Chinese, Danish). Papers with titles that seemed relevant but in languages that were non-comprehensible to the authors are listed in Appendix A (non-English studies).

### 2.2. Information Sources

Our literature search strategy was developed by using Medical Subject Headings (MeSH) and text words related to qSOFA. We searched the Cochrane Central Register of Controlled Trials (November 2016), EMBASE (1910 to Present), BIOSIS (2001 to 2012), OVID MEDLINE^®^ Epub Ahead of Print, In-Process & Other Non-Indexed Citations and OVID MEDLINE^®^ (1946 to Present with Daily Update), OVID Nursing Database (1946 to January Week 1 2017), and the Joanna Briggs Institute EBP Database, using the OVID interface. The WHO International Clinical Trial Registry Platform, Web of Science, Scopus, and ClinicalTrials.gov were searched independently.

### 2.3. Search Strategy

We have used the following terms to search ((((qSOFA) OR quick SOFA) OR quick sequential organ failure assessment) OR quick sepsis-related organ failure assessment) AND mortality.

Details may be found in Appendix B, Appendix C, Appendix D, Appendix E and Appendix F (search strategies).

### 2.4. Study Selection

Duplicates were removed, and records were identified and screened by LL and RL. After this, studies with no results available and studies in languages that our group could not read were also excluded. The remaining studies were discussed in a consensus meeting by CAG, MB, KH, LL, and RL. The results were compared at each stage, and discrepancies were discussed. If no consensus was met, CAG acted as the final adjudicator for the decision of whether a study should be included.

### 2.5. Data

Data was collected independently and was cross-checked by at least three reviewers. The data items extracted included study type (retrospective/prospective), sample size, patient characteristics such as age and gender, recruitment period, patient setting (location of recruitment), patient group (infection/‘all-comers’), mentation assessment, and the timing of qSOFA.

### 2.6. Outcomes

Our primary outcome was in-hospital mortality. Secondary outcomes were 1-month mortality, ICU admission, sepsis diagnosis, ICU length-of-stay, and hospital length-of-stay. We performed sub-group analyses for studies that only included patients with infection versus all-comers, the location of recruitment, altered mental status, and timing of qSOFA.

Graphs were generated using MedCalc Statistical Software version 18.11 [13].

### 2.7. Risk of Bias in Individual Studies

All studies included were assessed by using an adapted version of the Quality in Prognosis Studies instrument [14]. Six potential bias domains were explored: selection bias, bias in definition and measurement, outcome measurement bias, handling of missing data, confounding, and bias of statistics or the presentation of result. These six domains were be graded as “high risk (of bias)”, “low risk (of bias)”, or “unclear”.

Summary measures: The principal summary measure was the area under the receiver operator characteristic (AUROC) curve for the prediction of mortality. Sensitivity, specificity, positive predictive value (PPV), and negative predictive value (NPV) were also collected. All measures were also reported for Intensive Care Units (ICU) admission and sepsis diagnosis.

## 3. Results

### 3.1. Study Selection

The database search identified 529 records. After duplicates were removed, 251 records were identified and screened by LL and RL. After 117 abstracts were excluded, 24 ongoing trials with no results available, and seven records in languages that our group could not read were also excluded (all seven of these papers appeared to be reviews or articles that contained no original data). The remaining 103 were discussed in a consensus meeting by CAG, MB, KH, LL, and RL. We included 45 papers in the final analysis [15,16,17,18,19,20,21,22,23,24,25,26,27,28,29,30,31,32,33,34,35,36,37,38,39,40,41,42,43,44,45,46,47,48,49,50,51,52,53,54,55,56,57,58,59] (Figure 1). Excluded studies and the reasons for their exclusion are listed in Appendix G (Table A1).

### 3.2. Study and Sample Characteristics

Of the 45 studies, 27 were retrospective cohorts, 13 had data prospectively collected but retrospectively analyzed, and five were prospective cohorts. The studies recruited a total of 413,634 patients from Europe, North America, Asia, and Australasia, with a median age ranging from 49 to 80 years. Seven studies recruited patients from all settings, 24 studies recruited only ED patients, eight from ICU only, one from all non-ICU settings, one from general wards, one from a pre-hospital setting, and 13 included patients from more than one setting (e.g., ward, ICU, or ED). The recruitment periods ranged from one day (cross-sectional study) to 20 years (1996–2015). Sample sizes ranged from 58 to 184,875. Some 27 studies reported data on in-hospital mortality and 16 reported data on 1-month mortality (Table 1).

### 3.3. Risk of Bias within Studies

The individual assessments of risk of bias for the individual studies can be found in Appendix H.

“Selection bias” and “bias in definition” were the most common biases. The most noticeable inconsistency between all of the reviewed studies revolved around the definition of qSOFA. “Outcome measurement bias” was the least common bias (Table 2).

#### 3.3.1. Criteria of qSOFA

The original cut-off values for respiratory rate and systolic blood pressure were followed by most studies. There were large disagreements in the definitions of “altered mentation” between different papers. It was variously defined as different levels of the Glasgow Coma Scale (GCS); different levels of the AVPU (Alert, Pain, Voice, Unresponsive) scale, physician/nursing discretion, and even with more than one criterion being used in the same study, e.g., ‘GCS<14 or anything other than alert on the AVPU scale’.

#### 3.3.2. In-Hospital Mortality

From the 27 studies with a total of 380,041 patients that had data on in-hospital mortality, the median AUROC was 0.68, with a range from 0.55 to 0.82 (Figure 2). A total of 24 studies had data on sensitivity and specificity, ranging from 0.16 to 0.98 (median 0.52) and 0.19 to 0.97 (median 0.81), respectively. Positive and negative predictive values were reported in 18 studies with a range of 0.10–0.38 (median 0.2) and 0.85–0.99 (median 0.95), respectively. Positive and negative likelihood ratios were available in 12 studies, ranging from 1.2 to 4 (median 1.83), and 0.24 to 0.84 (median 0.59), respectively.

A high heterogeneity was confirmed by meta-analysis, with an I^2^ of 98.77%. A meta-analysis would therefore not yield meaningful results, with the data being extracted from these studies.

#### 3.3.3. Month (28/30 Day) Mortality

A total of 14 studies, with 35,775 patients reported 1-month mortality data (Figure 3). The median AUROC ranged from 0.58 to 0.85 (median 0.69). Sensitivity data were available in 12 of these studies, which ranged from 0.06 to 0.71 (median 0.43); specificity data were available in 13 studies, and ranged from 0.10 to 1.00 (median 0.84). PPV and NPV data were available in 10 studies, and they ranged from 0.14 to 0.68 (median 0.34) and 0.69 to 0.97 (median 0.91), respectively. Positive and negative likelihood ratio data were available in eight studies, and the values ranged from 1.99 to 4.66 (median 2.22) and 0.3 to 0.9 (median 6.43), respectively.

#### 3.3.4. ICU Admission

From the 12 studies that reported data on ICU admission, AUROC ranged from 0.58–0.81 (median 0.65, Figure 4. AUROC for ICU admission). Ten studies had data on sensitivity and specificity, which ranged from 0.1 to 0.74 (median 0.37) and 0.42 to 0.97 (median 0.86), respectively. The positive predictive value and negative predictive value data were 0.089–0.578 (median 0.38) in eight studies, and 0.19–0.99 (median 0.90) in nine studies, respectively. Positive and negative likelihood ratio data were available in eight studies, and ranged from 1.27 to 9.97 (median 2.68) and 0.5 to 0.9 (median 0.63), respectively.

#### 3.3.5. Hospital and ICU Length-of-Stay (LOS)

There were no studies that reported on the predicted ability of qSOFA for median ICU or hospital LOS. However, three studies that reported on median ICU LOS. Studies reported results that ranged from 2.9 to 3.1 days. Hospital LOS, presented in median time in qSOFA-positive patients were available in five studies, ranging from 5 to 15 days (a median of nine days).

#### 3.3.6. Diagnosis of Sepsis/Infection

Infective/septic diagnostic predictive values were only presented in two studies, Forward et al. [27] reported an AUROC for patients diagnosed with sepsis to be 0.88, and Brabrand et al. [19] reported an AUROC 0.88 for patients with a diagnosis of infection.

### 3.4. Summary of Results

Subgroup analyses of AUROC of in-hospital mortality were inconclusive. There was no obvious difference between location of patients who presented with or without infection (Appendix I/Figure A1), location of recruitment/data collection (Appendix J/Figure A2), how mentation was defined or measured (Appendix K/Figure A3), or the timing of qSOFA (Appendix L/Figure A4). A summary of the prognostic values reported from the studies reviewed may be found in Table 3.

## 4. Discussion

This systematic review of 45 studies with 413,634 patients showed that the AUROC of qSOFA for the in-hospital mortality in all patients (with or without suspected infection) was poor, and it showed that it was not suitable for routine clinical use. The AUROC values for other outcomes were also too low for qSOFA to be clinically useful.

qSOFA was developed to predict the likelihood of organ dysfunction in patients with suspected infection [50]. However, the detection of sepsis or infection may be clinically difficult, as symptoms of infection are highly variable [60], and they often mimic other diseases [61]. Misdiagnosis or late diagnosis have been associated with poorer outcomes [62]. Since diagnosis and detection may be difficult to achieve, screening for all patients and not just those with suspected infection would reduce subjectivity and avoidable error in the diagnostic process, and may be a better approach to reduce more severe outcomes and preventable deaths.

When initially introduced, qSOFA was reported to have an AUROC of 0.81 for predicting 1-month mortality. However, this value “was derived from models that include baseline variables plus candidate criteria” [50]. The candidate variables were age, Charlson comorbidity index, race/ethnicity, and gender. A subsequent comparison of the adjusted and unadjusted results in other studies showed that there were substantial differences between the two: Donnelly et al. adjusted 0.76 vs. unadjusted 0.66 [24]; Raith et al. adjusted 0.76 vs. unadjusted 0.61 [48]. We would therefore argue that the adjusted AUROC value reported by the original group bears little relevance for front-line clinicians.

Presenting prognostic predictions using AUROC has limitations [63], as it may be useful on a population scale, but it may not help clinicians on an individual level. In the emergency setting, high sensitivity is particularly important for supporting decisions for triage placement, and for screening and discharging patients; whereas specificity might be more relevant to the ward or ICU setting, to indicate whether a patient’s treatment should be escalated. The data obtained in this review showed the poor sensitivity and mediocre specificity of qSOFA for in-hospital mortality, 1-month mortality, and ICU admission. This suggests qSOFA’s poor utility for screening patients, and its modest value for escalation of care. The positive predictive values were also poor. Although the negative predictive values appeared to be good, the high negative predictive value is likely to reflect on the low incidence of the outcome measure.

The principal idea behind the development of qSOFA was to improve on the pre-existing Systemic Inflammatory Response Syndrome (SIRS) criteria for sepsis identification. Most studies that we reviewed showed that the AUROC for qSOFA outperforms SIRS for predicting in-hospital mortality. However, other scores such as the National Early Warning Score and the Modified Early Warning Score had been reported to have better prognostic values than both SIRS or qSOFA (NEWS 0.77, MEWS 0.73, qSOFA 0.69, and SIRS 0.65) [22]. All three scores had a higher sensitivity at their recommended cut off value when compared to qSOFA (SIRS 0.94, NEWS 0.86, MEWS 0.71, and qSOFA 0.69) [22]. Other systematic reviews focused on the comparison of qSOFA and SIRS, and on qSOFA as a prognostic tool in patients with suspected infection outside of ICU. All three reviews unanimously reported qSOFA’s poor sensitivity [64,65,66].

Two of the three variables in qSOFA are often measured and documented routinely. An assessment of mentation, however, requires experience and clinical judgment. The disagreements in the definition of “altered mentation” were a major source of bias, as they varied between different studies. In Seymour’s original qSOFA paper, the group reported that “the predictive validity of qSOFA was not significantly different when using … the GCS score <15 (*p* = 0.56), compared with the model with GCS score ≤13.” A standardized definition is required for future studies, and details must be added, to further elaborate on how altered mentation is determined in patients with impaired mental status at baseline, e.g., dementia sufferers. This is significant, as infection and sepsis are common causes of delirium in the older population.

The strengths of this review include the large number of study subjects, the inclusive search strategy, and bias assessment from multiple reviewers. However, there are also limitations to our review. We had taken a pragmatic approach in utilizing the qSOFA score, and we have used it on all-comers, rather than only on those with a suspected infection. Changes in treatment outcomes of sepsis made older studies difficult to compare directly with the more recent ones. The small number of prospective studies also limits the validity and generalizability of the results. There were only three prospective studies among the papers reviewed.

## 5. Conclusions

In conclusion, our group found that qSOFA is not a clinically useful prognostic tool for in-hospital, 1-month mortality, or ICU admission for all-comers, with or without suspected infection.

## Figures and Tables

**Figure 1 jcm-08-00061-f001:**
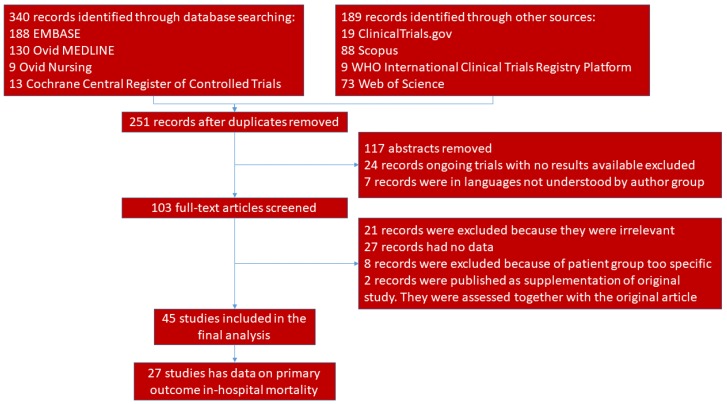
Study Flow.

**Figure 2 jcm-08-00061-f002:**
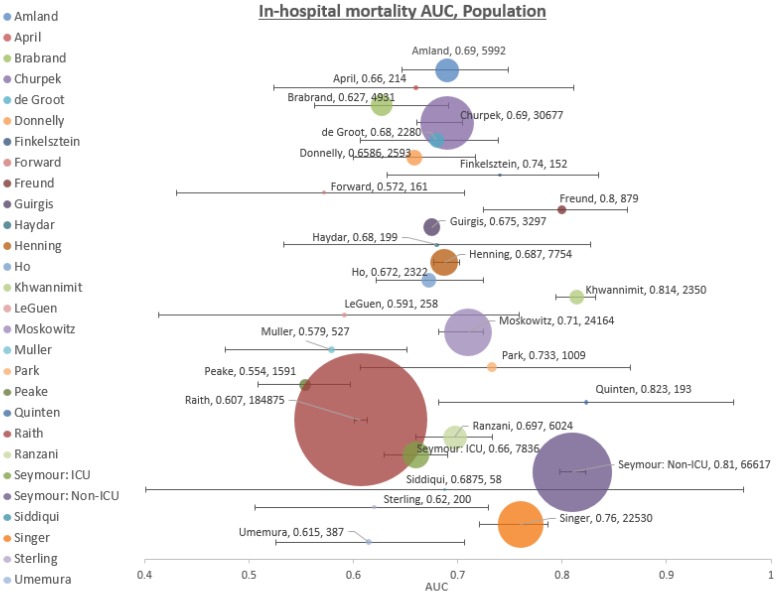
AUROC for in-hospital mortality.

**Figure 3 jcm-08-00061-f003:**
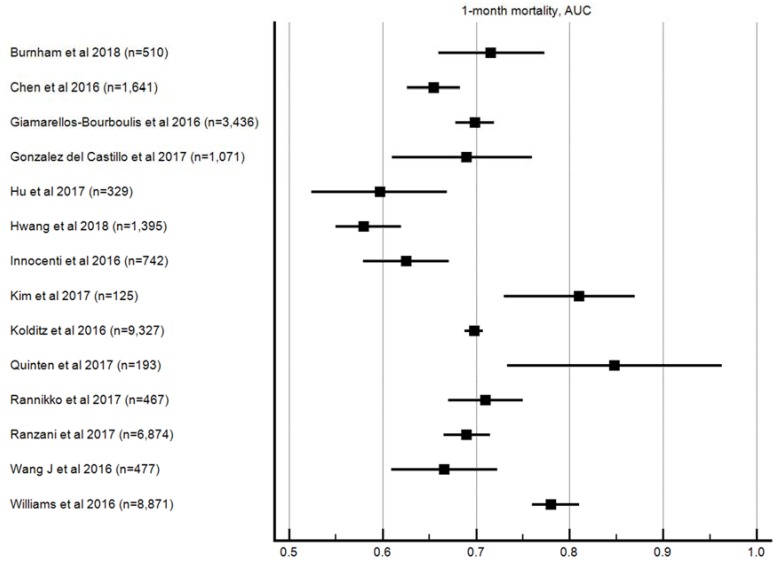
AUROC for 1-month mortality.

**Figure 4 jcm-08-00061-f004:**
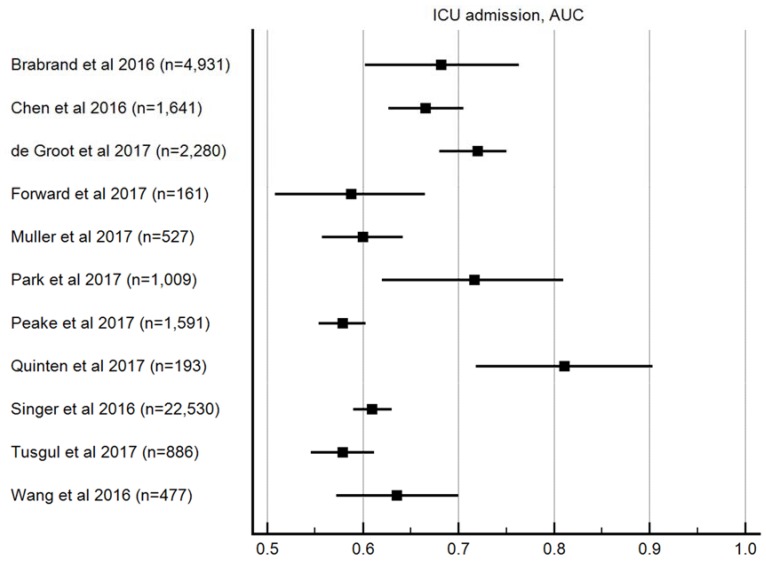
AUROC for ICU admission.

**Table 1 jcm-08-00061-t001:** Characteristics of the studies included in the systematic review of qSOFA for predicting prognosis.

Study	Median Age (IQR) Mean Age ± SD	Location	Male (%)	Sample Size	Study Type	Recruitment Period
Amland et al. [15]	65 (51–76)	US	48	5992	Retrospective	January 2016–March 2016
April et al. [16]	72 (60–79)	Texas, US	58.9	214	Retrospective	August 2012–February 2015
Askim et al. [17]	62 (41–78)	Norway	53	1535	PCDRC	January 2012–December 2012
Boulos et al. [18]	68.5 ± 17.4	Monash, Australia	52	646	Retrospective	January 2015–December 2015
Brabrand et al. [19]	65 (50–77)	Denmark	49.2	4931	Retrospective	October 2008–May 2010
Burnham et al. [20]	61.1 (51.6–69.8)	Missouri, USA	52	510	Retrospective	June 2009–December 2013
Chen et al. [21]	73 (62–79)	Beijing, China	59	1641	PCDRC	January 2012–May 2014
Churpek et al. [22]	58 ± 18	Chicago, US	47	30,677	Retrospective	November 2008–January 2016
de Groot et al. [23]	61.1 ± 17	Holland	57.7	2280	PCDRC	April 2011–February 2016
Donnelly et al. [24]	68 (61–75)	USA	47.8	2593	Retrospective	January 2003–October 2007
Du et al. [25]	56.4 ± 18.1	Sichuan, China	65.7	565	Retrospective	August 2015–July 2016
Finkelsztein et al. [26]	64 (51–75)	New York, USA	31	152	PCDRC	October 2014–July 2016
Forward et al. [27]	70 ± ?	Sydney, Australia	55	161	Prospective	May 2015–August 2015
Freund et al. [28]	67 (48–81)	Europe	53	879	Prospective	May 2016–June 2016
Giamarellos-Bourboulis et al. [29]	76 (IQR: 22)	Greece	?	3436	Retrospective	May 2006–December 2015
Gonzalez del Castillo et al. [30]	83.6 ± 5.6	Spain	50.8	1071	Prospective	October 2015–April 2016
Guirgis et al. [31]	59 (48–70)	Florida, USA	49	3297	Retrospective	October 2013–May 2016
Haydar et al. [32]	71 (range 18–102)	Portland, USA	55	199	Retrospective	September 2014–September 2015
Henning et al. [33]	58.4 ± 20.1	USA	52.2	7754	PCDRC	December 2003–September 2006
Ho et al. [34]	57.1 (41–70)	Perth, Australia	61	2322	PCDRC	January 2008–December 2013
Hu et al. [35]	?	Zhejiang, China	62.6	329	Retrospective	January 2015–June 2015
Hwang et al. [36]	65 (55–73)	Seoul, South Korea	56	1395	Retrospective	August 2008–September 2014
Innocenti et al. [37]	75 ± 14	Florence, Italy	53	742	Retrospective	June 2008–April 2016
Khwannimit et al. [38]	62 (45–75)	Songkhla, Thailand	56.1	2350	Retrospective	January 2007–December 2016
Kim et al. [39]	72 (59.5–80)	Seoul, South Korea	62.4	125	Retrospective	January 2014–December 2014
Kolditz et al. [40]	63 (?)	Germany	56	9327	Retrospective	October 2002–June 2015
LeGuen et al. [41]	72 (57–82)	Victoria, Australia	48	258	Prospective	6 June 2016, 10 July 2016
Moskowitz et al. [42]	63.8 ± 18.1	USA	50.9	24,164	Retrospective	January 2010–December 2014
Muller et al. [43]	66 (50–76)	Switzerland	64.5	527	Retrospective	June 2011–May 2013
Park et al. [44]	67.4 ± 17.6	Seoul, South Korea	45	1009	Retrospective	March 2007–February 2016
Peake et al. [45]	62.9 ± 16.5	Australasia	59.7	1591	PCDRC	October 2008–April 2014
Quinten et al. [46]	60 (48–71)	Netherlands	56	193	PCDRC	August 2012–April 2014
Raith et al. [47]	62.9 ± 17.4	Australasia	55.4	184,875	Retrospective	January 2000–December 2015
Rannikko et al. [48]	68 (58–78)	Finland	53	467	Retrospective	March 2012–February 2014
Ranzani et al. [49]	66.1 ± 19	Barcelona + Valencia, Spain	62.2	6874	PCDRC	January 1996–December 2015
Seymour et al. [50]	61 ± 19	US and Germany	43	74,453	Retrospective	January 2010–December 2012
Siddiqui et al. [51]	64.4 ± 12.9	Singapore	60	58	Retrospective	January 2015–December 2015
Singer et al. [52]	54 ± 21	New York, USA	47	200	Retrospective	January 2014–March 2015
Sterling et al. [53]	60 ± 16.7	USA	?	22,530	PCDRC	August 2004–January 2009
Szakmany et al. [54]	74 (61–83)	Wales, UK	47	380	Prospective	19 October 2016
Tusgul et al. [55]	80 (69–87)	Switzerland	52.1	886	Retrospective	January 2012–December 2012
Umemura et al. [56]	?	Japan	59.7	387	PCDRC	June 2010–May 2011
Wang J et al. [57]	73 (60–79)	Beijing, China	61.8	477	PCDRC	July 2015–December 2015
Wang S et al. [58]	63 ± 17.3	Chenzhou, China	69.5	311	Retrospective	July 2012–June 2016
Williams et al. [59]	49 (30–69)	Brisbane, Australia	51.3	8871	PCDRC	October 2007–May 2011

qSOFA, quick Sepsis-related Organ Failure Assessment; IQR, Interquartile Range; PCDRC, Prospectively Collected Data Retrospective Cohort; ?, Information not available.

**Table 2 jcm-08-00061-t002:** Risk of bias across the studies.

Author Year	Selection Bias	Bias in Definition and Measurement	Outcome Measurement Bias	Handling of Missing Data	Confounding	Bias of Statistics or Presentation of Result
Amland et al. 2017						
April et al. 2016						
Askim et al. 2017						
Boulos et al. 2017						
Brabrand et al. 2016						
Burnham et al. 2018						
Chen et al. 2016						
Churpek et al. 2017						
de Groot et al. 2017						
Donnelly et al. 2017						
Du et al. 2017						
Finkelsztein et al. 2017						
Forward et al. 2017						
Freund et al. 2016						
Giamarellos-Bourboulis et al. 2016						
Gonzalez del Castillo et al. 2017						
Guirgis et al. 2017						
Haydar et al. 2017						
Henning et al. 2017						
Ho et al. 2016						
Hu et al. 2017						
Hwang et al. 2018						
Innocenti et al. 2016						
Khwannimit et al. 2018						
Kim et al. 2017						
Kolditz et al. 2016						
LeGuen et al. 2017						
Moskowitz et al. 2017						
Muller et al. 2017						
Park et al. 2017						
Peake et al. 2017						
Quinten et al. 2017						
Raith et al. 2017						
Rannikko et al. 2017						
Ranzani et al. 2017						
Seymour et al. 2016						
Siddiqui et al. 2017						
Singer et al. 2016						
Sterling et al. 2017						
Szakmany et al. 2018						
Tusgul et al. 2017						
Umemura et al. 2017						
Wang J et al. 2016						
Wang S et al. 2017						
Williams et al. 2016						

Green, low risk; Yellow, moderate risk; Red, high risk.

**Table 3 jcm-08-00061-t003:** Summary of the prognostic values reported from the studies reviewed.

	qSOFA Median Value Min–Max (Number of Patients that the Value is Derived from)						
Outcomes	AUROC	Sensitivity	Specificity	PPV	NPV	LR+	LR−
In-hospital mortality	0.68	0.52	0.81	0.2	0.94	1.83	0.59
	0.55–0.82	0.16–0.98	0.19–0.97	0.07–0.38	0.85–0.99	1.15–4	0.24–0.84
	(*n* = 380,920)	(*n* = 118,051)	(*n* = 118,051)	(*n* = 67,555)	(*n* = 90,085)	(*n* = 24,925)	(*n* = 24,925)
1-month mortality	0.69	0.43	0.84	0.34	0.91	2.22	6.43
	0.58–0.85	0.06–0.71	0.10–1.00	0.14–0.68	0.69–0.97	1.26–3.71	2.17–14.4
	(*n* = 36,415)	(*n* = 34,462)	(*n* = 36,415)	(*n* = 26,603)	(*n* = 26,603)	(*n* = 8121)	(*n* = 8121)
ICU admission	0.65	0.37	0.86	0.38	0.9	2.68	0.63
	0.58–0.81	0.1–0.74	0.42–0.97	0.09–0.90	0.19–0.99	1.27–9.97	0.5–0.9
	(*n* = 37,105)	(*n* = 33,816)	(*n* = 33,816)	(*n* = 11,093)	(*n* = 33,623)	(*n* = 11,286)	(*n* = 11,286)

qSOFA, quick Sepsis-related Organ Failure Assessment; AUROC, Area Under the Receiver Operating Characteristics curve; PPV, Positive Predicted Value; NPV, Negative Predicted Value; LR+, Positive Likelihood Ratio; LR−, Negative Likelihood Ratio; ICU, Intensive Care Unit.

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
