# Peer review of "qSOFA is a Poor Predictor of Short-Term Mortality in All Patients: A Systematic Review of 410,000 Patients"

_jcm, 2019, doi:10.3390/jcm8010061_

Round 1
Reviewer 1 Report
The purpose of this paper was to assess the validity of qSOFA in the prediction of outcome (in‐hospital 12 and 1‐month mortality, ICU admission, and hospital and ICU length of stay) in patient without specific diagnosis. Taking into consideration that qSOFA was designed as a promising tool for quick and proper identification of patients at risk for sepsis, authors rationale is not consistent with the scope of using qSOFA, although studies that validate qSOFA are imperative.
Data provided are interesting but further analysis for patients with suspected infection should be also provided in order to compare the efficacy of using qSOFA between these subgroups. Otherwise data submitted does not add pivotal information to existing knowledge.
References must also be updated and extensive editing of English language is required.
In conclusion, this paper contains very interesting data although there is need for major revision.
Author Response
Data provided are interesting but further analysis for patients with suspected infection should be also provided in order to compare the efficacy of using qSOFA between these subgroups. Otherwise data submitted does not add pivotal information to existing knowledge.
Thank you for your comment. We do not find analyzing patients with suspected infection only relevant to our setting. Sepsis or suspected infection is difficult to identify (1, 2). Nevertheless we have analyzed the data from patients with suspected infection and it was included in appendix I of our article.
1. Alberto, L., Marshall, A. P., Walker, R., & Aitken, L. M. (2017). Screening for sepsis in general hospitalized patients: a systematic review. Journal of Hospital Infection, 96(4), 305-315.
2. Vincent, J. L. (2016). The clinical challenge of sepsis identification and monitoring. PLoS medicine, 13(5), e1002022.
References must also be updated and extensive editing of English language is required.
Thank you for your comment. We consider our references to be up-to-date for a systematic review. For language issues, this article was written by an authors who are fluent in English. Additionally, it was reviewed by an author whose native language is English.
In conclusion, this paper contains very interesting data although there is need for major revision.
Reviewer 2 Report
Lo et al. performed a meta-analysis of observational studies aiming to evaluate how qSOFA works in the prediction of outcome (in‐hospital 12 and 1‐month mortality, ICU admission, and hospital and ICU length of stay) in adult patient with sepsis. The paper is interesting, with some issues needed to be addressed
Abstract
Authors should add that patients evaluated presented with sepsis
Median AUC should be removed, and only the pooled one should be reported, along with pooled sensitivity and specificity
Methods
- Overall they are poorly written and should be rephrased.
- Statistical analysis in unclear: authors should add how they pooled aucs or sensitivity or specificity, the software used and the levels of heterogeneity assessed and the way they choose to manage it.
- Subgroup analysis for derivation vs. validation cohoorts should be added
Results
As in abstract, median AUC should be removed, and only the pooled one should be reported, along with pooled sensitivity and specificity
Tables should not be embedded in the text.
Author Response
Abstract
Authors should add that patients evaluated presented with sepsis
Thank you for your comment. This study aimed to explore the possibility for using qSOFA as a prognosticating tool for patients with and without infection/sepsis. Therefore studies reviewed include patients with and without infection.
Median AUC should be removed, and only the pooled one should be reported, along with pooled sensitivity and specificity
Thank you for your suggestion. Results of AUCs, sensitivity and specificity were extracted from all studies reviewed and they are presented in this review as ranges. The median value was derived.
Methods
- Overall they are poorly written and should be rephrased.
Thank you for your comment. We have reviewed the section and found it to accurately reflect how we have conducted this review. We have also reviewed other systematic reviews published by the JCM (1,2) and found that the main differences our manuscript had was the lack of subheadings. We have added these in.
1. Rubinkiewicz, M., Czerwińska, A., Zarzycki, P., Małczak, P., Nowakowski, M., Major, P., ... & Pędziwiatr, M. (2018). Comparison of Short-Term Clinical and Pathological Outcomes after Transanal versus Laparoscopic Total Mesorectal Excision for Low Anterior Rectal Resection Due to Rectal Cancer: A Systematic Review with Meta-Analysis. Journal of clinical medicine, 7(11), 448.
2. Legriel, S., & Brophy, G. (2016). Managing status epilepticus in the older adult. Journal of clinical medicine, 5(5), 53.
- Statistical analysis in unclear: authors should add how they pooled aucs or sensitivity or specificity, the software used and the levels of heterogeneity assessed and the way they choose to manage it.
Thank you for the suggestion. As mentioned above, results were not pooled. Software used for graph plotting and statistical analysis is MedCalc Statistical Software version 18.11 (MedCalc Software bvba, Ostend, Belgium; http://www.medcalc.org; 2018). Heterogeneity was high at 0.99 as reported.
- Subgroup analysis for derivation vs. validation cohoorts should be added
Thank you for your suggestion, however given that there is the original study by Seymour from which qSOFA was derived from, we decided not to compare their study with the rest of the studies.
Results
As in abstract, median AUC should be removed, and only the pooled one should be reported, along with pooled sensitivity and specificity
Thank you. Please see above
Tables should not be embedded in the text.
Thank you for your suggestion. We have inserted the table into the main text as per instructions of the journal. https://www.mdpi.com/journal/medicines/instructions#figures
Reviewer 3 Report
The authors present here a large systematic review evaluating the predictive ability of qSOFA for mortality in heterogeneous cohorts of patients. This article is affected by an "original sin" which is the potential biases derived from an incorrect design:
Major comments
- In line 41, the authors explain their rational for developing this work "Prior systematic reviews on the topic tend to focus on patients already identified as suspected infection. However, in an ED, the cause for attendance is not always clear and a diagnosis of infection is often made much later. We there believe that qSOFA should be applied earlier in the treatment process, before a specific condition is considered. This systemic review aims to determine the validity of qSOFA in the prediction of mortality in all patients, with or without suspected infection". This poses a potential major bias: qSOFA is designed to be used when the patient do have clinical suspicion of infection. The ability of qSOFA to predict mortality should be tested in cohorts of patients with clinical suspect of infection.
- In Methodology: the criteria for selecting studies to be considered in the review are not clear "We considered studies of all designs", please clarify.
- In Methodology authors say: "We found studies including patients presenting acutely to Emergency Departments and prehospital emergency care providers, critical care units (intensive care units and high dependency units) and general wards". In the results section:"Seven studies recruited patients from all settings, 24 studies recruited only ED patients, 8 from 120 ICU only, 1 from all non‐ICU settings, 1 from general wards, 1 from the pre‐hospital setting and 13 included patients from more than one setting (e.g. ward, ICU or ED). This means that authors mixed up patients at different moments of evolution, with distinct degree of severity...have the authors considered these factors as potential confusion variables in their analysis?
In conclusion, in this reviewer opinion the design of this study is not correct. If the authors want to test the ability of qSOFA in patients with no suspected infection, it would be better perform two separated analysis: one for patients with suspected infection and another one with no suspected infection. The diagnostic process of sepsis starts with the suspicion of infection. This cannot be ignored when qSOFA is tested. In addition, moment of disease evolution is also key: you cannot analyze the ability of qSOFA to predict mortality in patients admited to the ICU mixed with patients admitted to the ER....you should perform the analysis in homogeneous cohorts.
Minor comments:
- Title should reflect the kind of patients qSOFA is poor predictor of mortality for.
- Please correct "This systemic review" for "this systematic review"
Author Response
Major comments
- In line 41, the authors explain their rational for developing this work "Prior systematic reviews on the topic tend to focus on patients already identified as suspected infection. However, in an ED, the cause for attendance is not always clear and a diagnosis of infection is often made much later. We there believe that qSOFA should be applied earlier in the treatment process, before a specific condition is considered. This systemic review aims to determine the validity of qSOFA in the prediction of mortality in all patients, with or without suspected infection". This poses a potential major bias: qSOFA is designed to be used when the patient do have clinical suspicion of infection. The ability of qSOFA to predict mortality should be tested in cohorts of patients with clinical suspect of infection.
Thank you for your comments. The aim of this review is to provide a new angle of perspective from the Emergency Department’s point of view, particularly as the old SIRS criteria is no longer endorsed. Our group decided to include all patients (rather than only on those with suspected infection/sepsis) as sepsis can be difficult to diagnose in the ED.
- In Methodology: the criteria for selecting studies to be considered in the review are not clear "We considered studies of all designs", please clarify.
Thank you for your advice. It has been changed to “We considered studies of all designs, except for case series and case reports i.e. retrospective and prospective, observational and interventional studies. Studies only reported as abstracts were excluded.”
- In Methodology authors say: "We found studies including patients presenting acutely to Emergency Departments and prehospital emergency care providers, critical care units (intensive care units and high dependency units) and general wards". In the results section: "Seven studies recruited patients from all settings, 24 studies recruited only ED patients, 8 from ICU only, 1 from all non‐ICU settings, 1 from general wards, 1 from the pre‐hospital setting and 13 included patients from more than one setting (e.g. ward, ICU or ED). This means that authors mixed up patients at different moments of evolution, with distinct degree of severity...have the authors considered these factors as potential confusion variables in their analysis?
Thank you for your comment. Sub-analysis of location of recruitment has been included in Appendix J.
In conclusion, in this reviewer opinion the design of this study is not correct. If the authors want to test the ability of qSOFA in patients with no suspected infection, it would be better perform two separated analysis: one for patients with suspected infection and another one with no suspected infection. The diagnostic process of sepsis starts with the suspicion of infection. This cannot be ignored when qSOFA is tested. In addition, moment of disease evolution is also key: you cannot analyze the ability of qSOFA to predict mortality in patients admited to the ICU mixed with patients admitted to the ER....you should perform the analysis in homogeneous cohorts.
Whilst our group understand this reviewer’s sentiment, we disagree completely. From an Emergency medicine’s perspective, identifying patients with suspected infection or sepsis may be difficult and challenging (1,2). We have therefore taken a more pragmatic approach, including all patients, which reflects clinical reality.
1. Alberto, L., Marshall, A. P., Walker, R., & Aitken, L. M. (2017). Screening for sepsis in general hospitalized patients: a systematic review. Journal of Hospital Infection, 96(4), 305-315.
2. Vincent, J. L. (2016). The clinical challenge of sepsis identification and monitoring. PLoS medicine, 13(5), e1002022.
Minor comments:
- Title should reflect the kind of patients qSOFA is poor predictor of mortality for
Title has been changed to “qSOFA is a poor predictor of short term mortality in all patients: systematic review of 410,000 patients”
- Please correct "This systemic review" for "this systematic review"
Thank you for spotting this mistake. It has been changed.
Round 2
Reviewer 1 Report
Overall authors have made major improvement to manuscript providing valuable and convincing data. Present updated form provides novel information needed in order to validate qSOFA as a diagnosti tool. Data provided contribute to existing knowledge and underscore the limitations of qSOFA. It could be very interesting if we could have data from patients finally diagnosed with sepsis, however current paper version despite inherent limitations constitutes an updated and interesting scientific work.
Author Response
We thank you for your input and suggestions.
Reviewer 2 Report
the authors have fullfilled our requests
Author Response
Thank you for your kind review.
Reviewer 3 Report
The rational for the article design is still incorrect: "Prior systematic reviews on the topic tend to focus on patients already identified to have suspected infection. However, in an ED, the cause for attendance is not always clear and a diagnosis of infection is often made much later. We therefore believe that qSOFA should be applied earlier in the treatment process, before a specific condition is considered"
Again, qSOFA was proposed to detect sepsis in patients with suspected infection....if you are fair with the design of qSOFA, you have to test it in the conditions for what qSOFA was designed, which is presence of suspected infection.
In real life, you cannot use a predictor of sepsis mortality before infection is suspected. Moreover, the authors confound infection suspicion with infection diagnosis. This work present thus terrible methodological design flaws.
Otherwise, if you want to test the performance of qSOFA in all kind of patients, you have to be clear in the design of your work, not trying to link qSOFA with sepsis, as you actually do. Even in this scenario, you should design from the beginning separated analysis for qSOFA performance in patients with suspected sepsis and with no suspicion of sepsis.
The authors reply " From an Emergency medicine’s perspective, identifying patients with suspected infection or sepsis may be difficult and challenging. We have therefore taken a more pragmatic approach, including all patients, which reflects clinical reality." I strongly disagree...clinical reality for sepsis diagnosis starts with suspicion of infection...including all patients, even in absence of suspected infection, does not reflect clinical reality at all....
Author Response
Thank you again for this reviewer’s reply. We appreciate your input and standpoint however we would also like to re-iterate and emphasis that sepsis is a disease entity that is often difficult to diagnose, particularly in the Emergency department. Since SIRS has been superseded, clinicians will mistakenly use qSOFA the way SIRS was used. We believe that our paper highlights and confirms that qSOFA has a poor AUC for in-hospital mortality for all patients with or without suspicion of infection.
We may have to agree to disagree on this. If editors feel that this is a major flaw that will impede the chances of our article being published, please let us know.